# Exploiting Safe Spots in Neural Networks for Preemptive Robustness and Out-of-distribution Detection

## Abstract

Recent advances on adversarial defense mainly focus on improving the classifier's robustness against adversarially perturbed inputs. In this paper, we turn our attention from classifiers to inputs and explore if there exist *safe spots* in the vicinity of natural images that are robust to adversarial attacks. In this regard, we introduce a novel bi-level optimization algorithm that can find safe spots on over 90% of the correctly classified images for adversarially trained classifiers on CIFAR-10 and ImageNet datasets. Our experiments also show that they can be used to improve both the empirical and certified robustness on smoothed classifiers. Furthermore, by exploiting a novel safe spot inducing model training scheme and our safe spot generation method, we propose a new out-of-distribution detection algorithm which achieves the state of the art results on near-distribution outliers.

## 1 Introduction

Deep neural networks have achieved significant performance on various artificial intelligence tasks such as image classification, speech recognition, and reinforcement learning. Despite the results, Szegedy et al. (2013) demonstrated that deep neural networks are vulnerable to *adversarial examples*, minute input perturbations designed to mislead networks to yield incorrect predictions. There have been a large number of studies to improve the robustness of networks against adversarial perturbations (Song et al., 2017; Guo et al., 2018), while many of the proposed methods have been shown to fail against stronger adversaries (Athalye et al., 2018; Tramer et al., 2020). *Adversarial training* (Madry et al., 2017) and *randomized smoothing* (Cohen et al., 2019) are some of the few methods that survived the harsh verifications, each focusing on empirical and certified robustness, respectively.

To summarize, the study of adversarial examples has been an arms race between adversaries, who manipulate inputs to raise network malfunction, and defenders, who aim to preserve the network performance against the corrupted inputs. In this paper, we approach the adversarial robustness problem from a different perspective. Instead of defending networks from already perturbed examples, we assume the situation where the defenders can also influence inputs slightly for their interest before the adversaries' incursion. The defenders' goal for this manipulation will be to improve robustness by searching for spots in the input space that are resistant to adversarial attacks, given a pre-trained classifier. We explore methods for finding those *safe spots* from natural images under a given input modification budget and the degree of robustness achievable by utilizing these spots, which we denote as *preemptive robustness*. Ultimately, we tackle the following question:

- *Do safe spots always exist in the vicinity of natural images?*

One practical example of the proposed framework is the case where a user uploads his or her photo from local storage (*e.g.*, mobile device) to social media (*e.g.*, Instagram), as illustrated in Figure 1. Suppose there is an uploader (*A*) who posts a photo on social media, a web user (*B*) who queries a search engine (*e.g.*, Google) for an image, and a search engine that crawls images from social media, indexes them with a neural network, and retrieves the relevant images to *B*. Our threat model considers an adversary (*M*) that can download *A*'s image from social media, perturb it maliciously, and re-upload the perturbed image on the web, where the search engine may crawl and index images from. The classifier on the search engine will wrongly index the perturbed image, causing the search

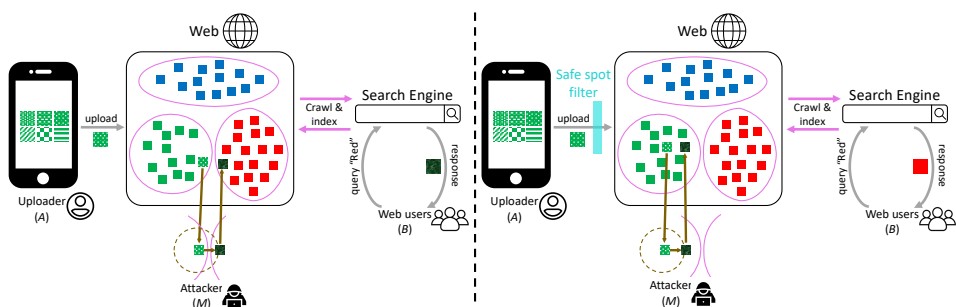

Figure 1: Overview of our proposed framework. The left side shows the web users retrieving wrong results due to the adversarial example. The right side adopts a safe spot filter on the image uploading process and succeeds in defending the query system from the attacker.

engine to malfunction. Suppose an African-American uploader (*A*) posts a photo of him or herself on social media, and a racist adversary (*M*) perturbs it to be misclassified as "gorilla" by the search engine. When another person (*B*) searches "gorilla" on Google, the perturbed image would appear, though the image content shows a photo of *A*. This attack fools both *A* and *B* since the perturbed image is used contrary to *A*'s purpose and is not the image *B* wanted. To prevent this, the social media company, cooperating with the search engine company, could ask if *A* agrees that the images are slightly changed when uploaded to make them robust to such attacks. The purpose of the modification process, corresponding to the "safe spot filter" in Figure 1, will be to ensure that the uploaded images are used under *A*'s intention and provide more accurate search results to *B*.

We develop a novel optimization problem for searching safe spots in the vicinity of natural images and observe that over 90% of the correctly classified images have safe spots nearby for adversarially trained models on both CIFAR-10 (Krizhevsky & Hinton, 2009) and ImageNet (Russakovsky et al., 2015). We also find that safe spots can enhance both empirical and certified robustness when applied on smoothed classifiers. Furthermore, we propose a novel safe spot inducing model training scheme to improve the preemptive robustness. By exploiting these *safe spot-aware* classifiers along with our safe spot search method, we also propose a new algorithm for *out-of-distribution detection*, which is often addressed together with robustness (Hendrycks et al., 2019a;c). Our algorithm outperforms other baselines on near-distribution outlier datasets such as CIFAR-100 (Krizhevsky & Hinton, 2009).

## 2    RELATED WORKS

**Adversarial training**    Goodfellow et al. (2015) first show that the robustness of a neural network can be enhanced by generating adversarial examples and including them in training set. PGD adversarial training improves the robustness against stronger adversarial attacks by augmenting training data with multi-step PGD adversarial examples (Madry et al., 2017). Some recent works report performance gains over PGD adversarial training by modifying the adversarial example generation procedure (Qin et al., 2019; Zhang & Wang, 2019; Zhang et al., 2020). However, most of the recent algorithmic improvements can be matched by simply using early stopping with PGD adversarial training (Rice et al., 2020; Croce & Hein, 2020). Other line of works achieve performance gains by utilizing additional datasets (Carmon et al., 2019; Wang et al., 2020; Hendrycks et al., 2019a).

**Randomized smoothing**    Injecting random noise during the forward pass can smooth the classifier's decision boundary and improve empirical robustness (Liu et al., 2018). Using differential privacy, Lecuyer et al. (2019) give theoretical guarantees for $\ell_1$ and $\ell_2$ robustness of classifiers smoothed with Gaussian and Laplacian noise. Cohen et al. (2019) provide a tight bound of $\ell_2$ robustness of networks smoothed with Gaussian noise via the Neyman-Pearson lemma. Another proof of the robustness bound was given in Salman et al. (2019) using Lipschitz property of smoothed classifiers, where they also propose a new adversarial training scheme for building robust smoothed classifiers.

**Out-of-Distribution detection with deep networks**    Although deep networks achieve high performance on various classification tasks, they also tend to yield high confidence in out-of-distribution samples (Nguyen et al., 2015). To filter out the anomalous examples, Hendrycks & Gimpel (2017) use the maximum value of a classifier's softmax distribution as a score function, while Lee et al. (2018)

propose Mahalanobis distance-based metric which spots out-of-distribution samples using hidden features. Hendrycks et al. (2019b) show that leveraging auxiliary datasets disjoint from test-time data can improve the detection performance. Recently, Sastry & Oore (2020) characterize activity patterns of hidden features by Gram matrices and use the matrix values to identify anomalies.

## 3 METHODS

### 3.1 GENERAL DEFINITION OF SAFE SPOT AND PREEMPTIVE ROBUSTNESS

We first establish a formal definition of *safe spot* and *preemptive robustness*. Let $c : \mathcal{X} \to \mathcal{Y}$ be a classifier which maps images to class labels. We define the *safe region* of the classifier $c$ as the set of images that $c$ can output robust predictions in the presence of slight adversarial perturbations.

**Definition 1** ($\epsilon$-safe region). *Let $c : \mathcal{X} \to \mathcal{Y}$ be a classifier and $\epsilon \in \mathbb{R}^+$ be the perturbation budget of an adversary. The $\epsilon$-safe region of the classifier $c$ is defined by $S_\epsilon(c) \coloneqq \{x \in \mathcal{X} \mid c(x') = c(x), \forall x' \in B_\epsilon(x)\}$.*

Here, $B_\epsilon(x) = \{x' \in \mathcal{X} \mid d(x, x') \le \epsilon\}$ is the $x$-centered $\epsilon$-ball. In this paper, we assume $\ell_p$ threat model, *i.e.*, $d(x, x') = \|x - x'\|_p$, which is the most common setting on adversarial robustness literature, and consider $p \in \{2, \infty\}$.

Now, suppose a defender can preemptively manipulate a natural image $x_o$ under a small modification budget, knowing its ground-truth label $y_o$. We denote the modified output image as $x_s$. Then, the defender's objective is to make $x_s$ be correctly classified as $y_o$ and locate in the safe region $S_\epsilon(c)$ to improve the robustness against adversarial attacks. If $x_s$ satisfies these two conditions, then we say $x_o$ is *preemptively robust* and $x_s$ is a *safe spot* of $x_o$.

**Definition 2** (Preemptive robustness). *Let $c : \mathcal{X} \to \mathcal{Y}$ be a classifier and $\delta, \epsilon \in \mathbb{R}^+$ be the modification budgets of the defender and the adversary, respectively. A natural image $x_o$ with its ground-truth label $y_o$ is called $(\delta, \epsilon)$-preemptively robust on the classifier $c$ if there exists a safe spot $x_s \in B_\delta(x_o)$ such that (i) $c(x_s) = y_o$ and (ii) $x_s \in S_\epsilon(c)$.*

### 3.2 SAFE SPOT SEARCH ALGORITHM

In this subsection, we develop an algorithm for searching a safe spot with a natural image. Given a classifier $c$, finding a safe spot $x_s$ from a natural image $x_o$ can be formulated as the following problem, which is directly from the definition of safe spot:

$$\underset{x_s}{\text{minimize}} \quad \mathbb{1}_{c(x_s) \ne y_o} + \mathbb{1}_{x_s \notin S_\epsilon(c)}$$
$$\text{subject to} \quad \|x_s - x_o\|_p \le \delta,$$

where $\mathbb{1}$ is the 0-1 loss function.

Note in this formulation the defender requires the ground-truth label $y_o$ for the safe spot search. However, images in the real-world (*e.g.*, social media) are usually unlabeled, unless uploaders annotate labels to their images by hand. So, it is natural to assume that the defender cannot access the ground-truth label $y_o$. In this case, we utilize the classifier's prediction $c(x_o)$ instead of $y_o$:

$$\underset{x_s}{\text{minimize}} \quad \mathbb{1}_{c(x_s) \ne c(x_o)} + \mathbb{1}_{x_s \notin S_\epsilon(c)}$$
$$\text{subject to} \quad \|x_s - x_o\|_p \le \delta.$$

As $x_s \notin S_\epsilon(c)$ implies there exists an adversarial example $x_a \in B_\epsilon(x_s)$ such that $c(x_a) \ne c(x_s)$, we can reformulate the optimization problem as

$$\underset{x_s}{\text{minimize}} \quad \mathbb{1}_{c(x_s) \ne c(x_o)} + \underset{x_a}{\sup} \; \mathbb{1}_{c(x_a) \ne c(x_s)}$$
$$\text{subject to} \quad \|x_s - x_o\|_p \le \delta \text{ and } \|x_a - x_s\|_p \le \epsilon.$$

Since the 0-1 loss is not differentiable, we employ the cross-entropy loss $\ell : \mathcal{X} \times \mathcal{Y} \to \mathbb{R}^+$ of the classifier $c$ as the convex surrogate loss function:

$$\underset{x_s}{\text{minimize}} \quad \ell(x_s, c(x_o)) + \underset{x_a}{\sup} \; \ell(x_a, c(x_s)) \tag{1}$$
$$\text{subject to} \quad \|x_s - x_o\|_p \le \delta \text{ and } \|x_a - x_s\|_p \le \epsilon.$$

Let $h(x_s)$ denote the objective in Equation (1). Instead of minimizing $h(x_s)$, we minimize $\tilde{h}(x_s) = \sup_{x_a} \ell(x_a, c(x_o))$, since it upper bounds $h(x_s)$ when sufficiently minimized by Lemma 1.

**Algorithm 1** Finding a safe spot

---

**input** An image and its prediction $(x_o, c(x_o))$, the cross-entropy
function $\ell$

$x_s = x_o$
**for** $i = 1, \ldots, \text{MAXITER}$ **do**
    Generate $N$ adversarial examples
    **for** $n = 1, \ldots, N$ **do**
        $x_{a,n}^{(0)} = x_s + \eta_n$ where $\eta_n \sim \mathcal{U}(B_\epsilon(0))$
        **for** $t = 1, \ldots, T$ **do**
            $x_{a,n}^{(t)} = \Pi_{x_s,\epsilon}\big(f(x_{a,n}^{(t-1)}; c(x_o), \ell)\big)$
        **end for**
    **end for**
    $x_s \leftarrow \Pi_{x_o,\delta}\left(x_s - \beta \cdot \dfrac{1}{N}\sum_{n=1}^{N}\dfrac{\partial \ell(x_{a,n}^{(T)}, c(x_o))}{\partial x_s}\right)$
**end for**
**output** $x_s$

---

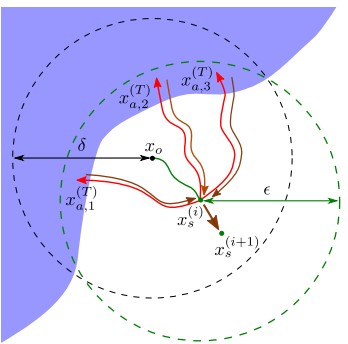

Figure 2: Illustration of the safe spot search process. The shaded region represents the set of points that are misclassified.

**Lemma 1.** *If* $\tilde{h}(x_s) \leq -\log(0.5) \simeq 0.6931$, *then* $h(x_s) \leq 2\tilde{h}(x_s)$.

*Proof.* See Supplementary A.1. □

Finally, we have the following optimization problem:

$$
\begin{aligned}
&\underset{x_s}{\text{minimize}} \quad \underset{x_a}{\sup} \; \ell(x_a, c(x_o)) \\
&\text{subject to } \|x_s - x_o\|_p \leq \delta \text{ and } \|x_a - x_s\|_p \leq \epsilon.
\end{aligned}
\tag{2}
$$

To solve Equation (2), we first approximate the inner maximization problem by running $T$-step PGD (Madry et al., 2017) whose dynamics is given by

$$
\begin{aligned}
x_a^{(0)} &= x_s + \eta && \text{(random start)} \\
\tilde{x}_a^{(t)} &= f\left(x_a^{(t-1)}; c(x_o), \ell\right) && \text{(adversarial update)} \\
x_a^{(t)} &= \Pi_{x_s,\epsilon}\left(\tilde{x}_a^{(t)}\right), && \text{(projection)}
\end{aligned}
$$

where $\eta$ is a random noise uniformly sampled from the $\ell_p$ zero-centered $\epsilon$-ball, $f$ is FGSM (Goodfellow et al., 2015) defined by

$$
f(x; y, \ell) = \begin{cases} x + \alpha \cdot \text{sgn}\left(\nabla_x \ell(x, y)\right) & \text{if } p = \infty \\ x + \alpha \cdot \dfrac{\nabla_x \ell(x, y)}{\|\nabla_x \ell(x, y)\|_2} & \text{if } p = 2, \end{cases}
$$

and $\Pi_{x_s,\epsilon}$ is a projection operation to $B_\epsilon(x_s)$. Then, we iteratively solve the approximate problem given by replacing $x_a$ to $x_a^{(T)}$ in Equation (2). To update $x_s$, we need to compute the gradient of $\ell(x_a^{(T)}, c(x_o))$ with respect to $x_s$ expressed as

$$
\frac{\partial \ell\left(x_a^{(T)}, c(x_o)\right)}{\partial x_s} = \frac{\partial \tilde{f}\left(x_a^{(0)}\right)^{\mathsf{T}}}{\partial x} \cdots \frac{\partial \tilde{f}\left(x_a^{(T-1)}\right)^{\mathsf{T}}}{\partial x} \cdot \nabla_x \ell\left(x_a^{(T)}, c(x_o)\right),
$$

where $\partial \tilde{f}/\partial x$ is the Jacobian matrix of $\tilde{f} = \Pi_{x_s,\epsilon} \circ f$ which is easily computed via back-propagation. After computing the gradient, we update $x_s$ by projected gradient descent method:

$$
x_s^{(i+1)} = \Pi_{x_o,\delta}\left(x_s^{(i)} - \beta \cdot \frac{\partial \ell(x_a^{(T)}, c(x_o))}{\partial x_s}\right).
$$

Note that the loss $\ell(x_a^{(T)}, c(x_o))$ is now a random variable dependent on $\eta$. Therefore, we generate $N$ adversarial examples $\{x_{a,n}^{(T)}\}_{n=1}^{N}$ with different noises and optimize the sample mean of the losses instead. Algorithm 1 shows the overall safe spot search algorithm and Figure 2 illustrates our optimization process.

### 3.3 COMPUTING UPDATE GRADIENT WITHOUT SECOND-ORDER DERIVATIVES

Computing the update gradient with respect to $x_s$ involves the use of second-order derivatives of the loss function $\ell$ since the dynamics $f$ contains the loss gradient $\nabla_x \ell(x, y)$. Standard deep learning libraries, such as PyTorch (Paszke et al., 2019), support the computation of higher-order derivatives. However, it imposes a huge memory burden as the size of the computational graph increases. Furthermore, for the case of $p = 2$, computing the update gradient with the second-order derivatives might cause *exploding gradient problem* if the loss gradient vanishes by Proposition 1.

**Lemma 2.** *Suppose $\ell$ is twice-differentiable and its second partial derivatives are continuous. If $p = 2$, the Jacobian of the dynamics $f$ is*

$$\frac{\partial f}{\partial x} = I + \alpha \cdot \left( I - \left( \frac{g}{\|g\|_2} \right) \left( \frac{g}{\|g\|_2} \right)^{\mathsf{T}} \right) \frac{H}{\|g\|_2},$$

*where $g = \nabla_x \ell(x, y)$ and $H = \nabla_x^2 \ell(x, y)$.*

*Proof.* See Supplementary B.1. □

**Proposition 1.** *If the maximum eigenvalue of $H$ in absolute value is $\sigma$, then*

$$\left\| \frac{\partial f}{\partial x}^{\mathsf{T}} \cdot a \right\|_2 \leq \left( 1 + \alpha \cdot \frac{\sigma}{\|g\|_2} \right) \|a\|_2.$$

*Proof.* See Supplementary B.2. □

As we update $x_s$, the loss gradients of $x_s$ and its adversarial examples $x_a$ get reduced to zero, which might cause the update gradient to explode and destabilize the update process. To address this problem, we approximate the update gradient by excluding the second-order derivatives, following the practice in Finn et al. (2017). We also include an experiment in comparison to using the exact update gradient in supplementary B.3. For the case of $p = \infty$, the second-order derivatives naturally vanish since we take the sign on the loss gradient $\nabla_x \ell(x, y)$. Therefore, the approximate gradient is equal to the exact update gradient.

### 3.4 FINDING A SAFE SPOT FOR CLASSIFIERS WITH RANDOMIZED SMOOTHING

To further enhance the robustness of our safe spot framework, we can leverage the randomized smoothing technique along with our algorithm. Given a base classifier $c : \mathcal{X} \to \mathcal{Y}$, the smoothed classifier $g : \mathcal{X} \to \mathcal{Y}$ is defined by

$$g(x) = \operatorname*{argmax}_{y \in \mathcal{Y}} \mathbb{P}\left(c(x + \eta) = y\right),$$

where $\eta \sim \mathcal{N}(0, \sigma^2 I)$. To find a safe spot $x_s$ of a natural image $x_o$, we have to find an adversarial example $x_a$ of $x_s$ that maximizes the cross-entropy loss $\ell(x_a, c(x_o))$ for solving the inner maximization problem in Equation (2). However, crafting adversarial examples for the smoothed classifier is ill-behaved since the argmax is non-differentiable. To address the problem, we follow the approach in Salman et al. (2019) and approximate the smoothed classifier $g$ with the smoothed soft classifier $G : \mathcal{X} \to P(\mathcal{Y})$ defined as

$$G(x) = \mathbb{E}_{\eta \sim \mathcal{N}(0, \sigma^2 I)} \left[ C(x + \eta) \right],$$

where $P(\mathcal{Y})$ is the set of probability distribution over $\mathcal{Y}$ and $C : \mathcal{X} \to P(\mathcal{Y})$ is the soft version of the base classifier $c$ such that $\operatorname{argmax}_{y \in \mathcal{Y}} C(x)_y = c(x)$. Finally, the adversarial example $x_a$ is found by maximizing the cross-entropy loss of $G$ instead:

$$\operatorname*{maximize}_{x_a} \ -\log\left( G(x_a)_{c(x_o)} \right) \tag{3}$$

$$\text{subject to } \|x_a - x_s\|_p \leq \epsilon,$$

which can be approximated by $T$-step randomized PGD with $M$ restarts, where random noises are sampled from Gaussian distribution to compute the sample mean of the objective at each step. By replacing the inner maximization problem in Equation (2) by the randomized PGD, we can update $x_s$ similarly.

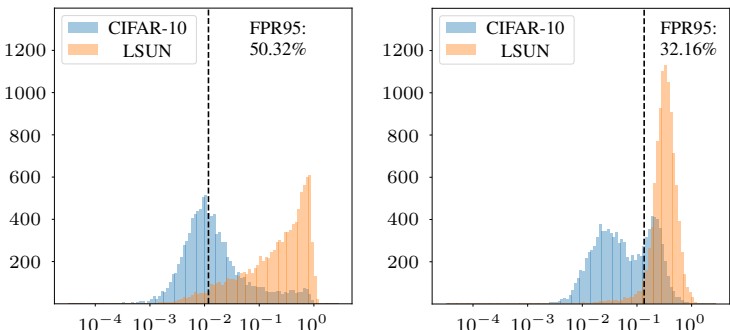

Figure 3: Histograms for the loss values of images $\ell(x_o, c(x_o))$ (left) and the loss values of the perturbed safe spot solution $\sup_{x_a^* \in B_\epsilon(x_s^*)} \ell(x_a^*, c(x_o))$ (right). A safe spot-aware adversarially trained model without fine-tuning is used as the classifier. The dotted lines are where the false positive rate is 95%. Detailed settings in Supplementary C.3.

## 3.5 SAFE SPOT-AWARE ADVERSARIAL TRAINING

In Section 3.2, we investigated how the defender can find a safe spot from a natural image, given a pre-trained classifier. In this subsection, we explore the defender's training scheme for a classifier on which data points are preemptively robust. Suppose the defender has a labeled training set, which is drawn from a true data distribution $\mathcal{D}$. To induce a classifier to have safe spots in the vicinity of data points, the defender's optimal training objective should have the following form:

$$\underset{\theta}{\text{minimize}} \quad \underset{(x_o, y_o) \sim \mathcal{D}}{\mathbb{E}} \left[ \ell(x_a^*, y_o; \theta) \right]$$
$$\text{subject to} \quad x_a^* = \underset{x_a \in B_\epsilon(x_s^*)}{\text{argmax}} \, \ell(x_a, y_o) \text{ and } x_s^* = \underset{x_s \in B_\delta(x_o)}{\text{argmin}} \, \underset{x_a \in B_\epsilon(x_s)}{\text{sup}} \, \ell(x_a, y_o),$$

where $\theta$ is the set of trainable parameters. Concretely, the defender finds a safe spot candidate $x_s^*$ of a datapoint $x_o$ and generates an adversarial example $x_a^*$ from $x_s^*$. Then, the defender minimizes the cross-entropy loss $\ell(x_a^*, y_o; \theta)$ so that $x_s^*$ becomes an actual safe spot. Note that the ground-truth label $y_o$ is used instead of the prediction $c(x_o)$, since we assume that the defender can access the ground-truth label during training.

The most direct way to optimize the objective would be to find $x_s^*$ from $x_o$ using our safe spot search algorithm and perform $k$-step PGD adversarial training (Madry et al., 2017) with $x_s^*$. However, since the safe spot search algorithm requires running $T$-step PGD dynamics per each update, the proposed training procedure would be more computationally demanding than PGD adversarial training. To ease this problem, we consider replacing the inner maximization $\sup_{x_a} \ell(x_a, y_o)$ in the safe spot search by $\ell(x_s, y_o)$:

$$x_s^* = \underset{x_s \in B_\delta(x_o)}{\text{argmin}} \, \underset{x_a \in B_\epsilon(x_s)}{\text{sup}} \, \ell(x_a, y_o) \implies x_s^* = \underset{x_s \in B_\delta(x_o)}{\text{argmin}} \, \ell(x_s, y_o).$$

Then, $x_s^*$ can be easily computed by running targeted FGSM or $k$-step PGD on $x_o$ towards the ground-truth label $y_o$. We denote this training scheme as *safe spot-aware adversarial training*.

## 3.6 OUT-OF-DISTRIBUTION DETECTION

The safe spot-aware adversarial training method induces the learned data distribution to have safe spots near its data points. Thus, we can naturally conjecture that the samples from the learned distribution will have a higher probability of having safe spots compared to the out-of-distribution (OOD) samples, as shown in Figure 3. We leverage this conjecture to propose a new out-of-distribution detection algorithm that jointly utilizes our safe spot generation method and safe spot-aware adversarial training.

Following the framework of Hendrycks et al. (2019b), which use auxiliary outlier data to tune anomaly detectors, we consider there are three types of data distributions, $\mathcal{D}_{\text{in}}$, $\mathcal{D}_{\text{out}}^{\text{train}}$, and $\mathcal{D}_{\text{out}}^{\text{test}}$. $\mathcal{D}_{\text{in}}$ refers to the learned distribution, also called the *in-distribution*. $\mathcal{D}_{\text{out}}^{\text{train}}$ is the given distribution of outliers used to tune the detection algorithm, which is orthogonal to $\mathcal{D}_{\text{out}}^{\text{test}}$. $\mathcal{D}_{\text{out}}^{\text{test}}$ is the distribution we want to detect as OOD during inference, which is unknown. We include the auxiliary outlier data to

our safe spot-ware training procedure and adapt the training objective as below:

$$\underset{\theta}{\text{minimize}} \ \underset{(x_o, y_o) \sim \mathcal{D}_{\text{in}}}{\mathbb{E}} [\ell(x_a^*, y_o; \theta)] + \underset{\hat{x}_o \sim \mathcal{D}_{\text{out}}^{\text{train}}}{\mathbb{E}} [\gamma \cdot D_{\text{KL}}(\bar{y} \parallel C(\hat{x}_o; \theta)) - \lambda \cdot \ell(\hat{x}_a^*, c(\hat{x}_o); \theta)] \quad (4)$$

$$\text{subject to} \quad x_a^* = \underset{x_a \in B_\epsilon(x_s^*)}{\text{argmax}} \ \ell(x_a, y_o) \ \text{ and } \ x_s^* = \underset{x_s \in B_\delta(x_o)}{\text{argmin}} \ \underset{x_a \in B_\epsilon(x_s)}{\text{sup}} \ \ell(x_a, y_o)$$

$$\hat{x}_a^* = \underset{\hat{x}_a \in B_\epsilon(\hat{x}_s^*)}{\text{argmax}} \ \ell(\hat{x}_a, c(\hat{x}_o)) \ \text{ and } \ \hat{x}_s^* = \underset{\hat{x}_s \in B_\delta(\hat{x}_o)}{\text{argmin}} \ \underset{\hat{x}_a \in B_\epsilon(\hat{x}_s)}{\text{sup}} \ \ell(\hat{x}_a, c(\hat{x}_o)),$$

where $\bar{y}$ is the uniform distribution and $C(\hat{x}_o; \theta)$ is the softmax probability of $\hat{x}_o$. Since $\hat{x}_o$ is unlabeled, we use the prediction $c(\hat{x}_o)$ instead for safe spot searching. Note that if $\epsilon \geq \delta$, the first term in Equation (4) also maximizes the confidence of the original in-distribution samples, since $x_o \in B_\delta(x_s^*) \subseteq B_\epsilon(x_s^*)$ and therefore $\ell(x_o, y; \theta) \leq \ell(x_a^*, y; \theta)$. Similarly, the second and third terms minimize the prediction confidence and the probability of safe spot existence of the outlier samples, respectively.

With the trained classifier, we measure the safe spot objective value from Equation (2) along with the maximum softmax probability (MSP) and use the values as indicators to detect OOD samples. Concretely, we define the score function as a linear combination of the two indicators. Considering they have a different range of possible values, we replace the safe spot objective value with the MSP of the adversarial example for the safe spot solution. Finally, the score function is formulated as

$$D(x_o) := \mu \cdot \max_{y \in \mathcal{Y}} C(x_o)_y + (1 - \mu) \cdot \max_{y \in \mathcal{Y}} C(x_a^*)_y,$$

where $x_s^* \in B_\delta(x_o)$ is the optimal solution of the safe spot algorithm for $x_o$ and $x_a^* \in B_\epsilon(x_s^*)$ is the adversarial example of $x_s^*$. We filter inputs with low scores as OOD.

## 4 EXPERIMENTS

As it is natural to assume that the defender and the adversary have the same modification budget, we set $\delta = \epsilon$ for all experiments. We evaluate our methods by measuring clean and adversarial accuracies, where adversarial accuracy refers to the prediction score under 20-step untargeted PGD attack with a step size of $\epsilon/4$. In the experiment tables, **None** column indicates using original images as inputs, and **S-Full** uses safe spot images from Algorithm 1. We also evaluate safe spot search via targeted FGSM and 20-step PGD towards the class inferred from the classifier, each denoted as **S-FGSM** and **S-PGD**. Detailed settings are listed on supplementary C.

### 4.1 CIFAR-10

We use Wide-ResNet-34-10 (Zagoruyko & Komodakis, 2016) and consider two threat models, $\ell_\infty$ with $\epsilon = 8/255$ and $\ell_2$ with $\epsilon = 0.5$. We run our experiments on four differently trained models. The natural model is trained in a standard manner without considering adversaries. ADV is a PGD adversarially trained model. **S-FGSM**+ADV and **S-PGD**+ADV are safe spot-aware adversarially trained models, with safe spot search approximated by FGSM or 10-step PGD with a step size of $\delta/4$.

The $\ell_\infty$ threat model result in Table 1 (left) shows our methods can find safe spots on over $85\%$ of the test set images, except for the natural model. This performance is near the upper bound, which is the classifier's clean accuracy since we use predicted labels for safe spot search. We also observe that safe spot search via targeted FGSM or PGD is also feasible for ADV, S-FGSM+ADV, and S-PGD+ADV models, but they still miss on about $10\%$ of correctly classified images. When jointly used with our safe spot search method, the safe spot-aware training achieves the highest adversarial accuracy, along with a clean accuracy much higher than PGD adversarial training.

The $\ell_2$ threat model result in Table 1 (right) shows similar results as the $\ell_\infty$ experiment, except that the adversarial accuracy of safe spots generated by S-Full on the natural model is much higher. However, we note that the adversarial accuracy of safe spots on the natural model may go down to about $20\%$ when the attack gets stronger, for example, by increasing PGD iterations. The results on stronger PGD attacks and other types of attacks are considered in supplementary D.3 and D.4.

| Model | Method | | | | Model | Method | | | |
|---|---|---|---|---|---|---|---|---|---|
| | None | S-FGSM | S-PGD | S-Full | | None | S-FGSM | S-PGD | S-Full |
| Natural | 95.97/00.00 | 82.40/00.00 | 95.97/00.00 | 95.48/09.67 | Natural | 95.97/00.53 | 94.61/00.79 | 95.97/00.38 | 95.94/59.00 |
| ADV | 86.51/47.21 | 86.51/77.08 | 86.51/71.22 | 86.51/85.06 | ADV | 90.26/68.16 | 90.26/88.82 | 90.26/86.28 | 90.26/89.99 |
| **S-FGSM**+ADV | 86.83/42.50 | 86.83/78.23 | 86.83/69.25 | 86.83/85.35 | **S-FGSM**+ADV | 90.92/63.27 | 90.92/88.82 | 90.92/84.92 | 90.92/90.60 |
| **S-PGD**+ADV | 91.32/39.33 | 91.32/77.01 | 91.32/63.94 | 91.32/**89.84** | **S-PGD**+ADV | 94.10/57.70 | 94.10/88.03 | 94.10/80.94 | 94.10/**93.54** |

Table 1: Classification accuracy under $\ell_\infty$ threat with $\epsilon = 8/255$ (left), $\ell_2$ threat with $\epsilon = 0.5$ (right), on CIFAR-10. (clean acc./adv acc.)

## 4.2 IMAGENET

We use ResNet-50 and consider three threat models: $\ell_\infty$ with $\epsilon \in \{4/255, 8/255\}$ and $\ell_2$ with $\epsilon = 3.0$. For safe spot-aware adversarial training experiments, we utilize "fast" adversarial training (Wong et al., 2020) and train the safe spot-aware model S-FGSM+Fast to reduce the training cost.

Table 2 (left) shows results on $\ell_\infty$ attack under $\epsilon = 4/255$. Similar to results on CIFAR-10, our methods are capable of finding safe spots near to original images that are correctly classified on the robust classifiers. Also, our proposed safe spot-aware classifier outperforms the original robust classifier by a large margin in both clean and adversarial accuracies. Table 2 (right) shows results on $\ell_\infty$ on $\epsilon = 8/255$. In this setting, we also apply our algorithm to the ADV model trained with $\epsilon_{\text{train}} = 4/255$. Note that by changing only the $\epsilon_{\text{train}}$ value on adversarial training, we get a 10% gain on our safe spot's adversarial accuracy. Surprisingly, classifiers adversarially trained with smaller $\epsilon_{\text{train}}$ performs substantially better in terms of preemptive robustness compared to using more robust classifiers. This implies that the conventional notion of robustness does not necessarily translate to preemptive robustness. Experiments on $\ell_2$ attacks show similar results and can be found in Supplementary D.1.

| Model | Method | | | | Model | Method | | | |
|---|---|---|---|---|---|---|---|---|---|
| | None | S-FGSM | S-PGD | S-Full | | None | S-FGSM | S-PGD | S-Full |
| Natural | 75.63/00.03 | 74.87/00.47 | 75.52/00.27 | 75.63/08.22 | Natural | 75.63/00.01 | 73.05/00.09 | 75.04/00.18 | 75.54/02.40 |
| ADV | 61.35/32.57 | 61.35/56.50 | 61.35/53.13 | 61.35/60.06 | ADV ($\epsilon_{\text{train}} = 8/255$) | 47.11/18.35 | 47.05/39.04 | 47.11/37.83 | 47.08/45.42 |
| Natural | 70.81/00.01 | 70.33/00.21 | 70.79/00.18 | 70.82/07.41 | ADV ($\epsilon_{\text{train}} = 4/255$) | 61.35/11.64 | 61.36/32.44 | 61.35/28.58 | 61.35/**55.85** |
| Fast | 57.05/29.97 | 57.07/50.97 | 57.05/50.20 | 57.04/56.26 | | | | | |
| **S-FGSM**+Fast | 64.67/14.51 | 64.73/42.47 | 64.67/34.53 | 64.67/**61.97** | | | | | |

Table 2: Classification accuracy under $\ell_\infty$ threat with $\epsilon = 4/255$ (left) and $\epsilon = 8/255$ (right) on ImageNet. The lower three models on $\epsilon = 4/255$ are trained in Fast style. (clean acc./adv acc.)

## 4.3 RANDOMIZED SMOOTHING

We also evaluate our algorithm for classifiers with randomized smoothing. Here, we consider the $\ell_2$ threat model, where $\epsilon = 0.5$ for CIFAR-10 and $\epsilon = 3.0$ for ImageNet. We run experiments on the smoothed classifiers based on the natural and the Gaussian-noise augmented model, which are considered certifiably robust (Lecuyer et al., 2019; Cohen et al., 2019). We measure empirical robustness against the randomized PGD on both models and measure certified robustness on the Gaussian model. Detailed settings such as noise level $\sigma$ and randomized PGD are listed in supplementary C.2.

Table 3 shows our results on the empirical robustness against the randomized PGD. We observe that our algorithm can find safe spots for the natural model with randomized smoothing on 57% and 37% of correctly classified images of CIFAR-10 and ImageNet, respectively. Furthermore, as shown in supplementary D.3, the adversarial accuracy of the smoothed natural model does not suffer from accuracy drop when the attack becomes stronger, in contrast to the natural model. Also, the smoothed Gaussian model, whose training cost is comparable to standard training much less than PGD adversarial training, achieves higher clean and adversarial accuracy compared to the ADV model. Certified robustness results of smoothed classifiers can be found in Supplementary D.2, where our safe spot algorithm also improves the certified robustness on both the datasets.

| Model | Method | | Model | Method | |
| | None | S-Full | | None | S-Full |
| --- | --- | --- | --- | --- | --- |
| Natural | 95.97 / 00.53 | 95.94 / 59.00 | Natural | 75.63 / 00.01 | 75.63 / 10.14 |
| Natural+Smoothing | 72.39 / 03.31 | 94.92 / 55.02 | Natural+Smoothing | 48.93 / 00.50 | 74.76 / 27.84 |
| Gaussian+Smoothing | 92.35 / 56.03 | 92.30 / **91.35** | Gaussian+Smoothing | 69.90 / 10.03 | 70.03 / **62.78** |

Table 3: Empirical robustness of randomized smoothed networks under $\ell_2$ threat with $\epsilon = 0.5$ on CIFAR-10 (left) and with $\epsilon = 3.0$ on ImageNet (right). (clean acc./adv acc.)

## 4.4 OUT-OF-DISTRIBUTION DETECTION

We evaluate the performance of our proposed detection algorithm on models trained with CIFAR-10. We consider various OOD datasets including CIFAR-100, SVHN (Netzer et al., 2011), TinyImageNet (Johnson et al.), LSUN (Yu et al., 2015), and synthetic noise. Following the experimental protocol of Hendrycks et al. (2019b), we evaluate the OOD detection methods on three metrics: area under the receiver operating characteristic curve (**AUROC**), area under the precision-recall curve (**AUPR**), and the false positive rate at 95% true positive rate (**FPR95**). We compare our method's performance to Mahalanobis (Lee et al., 2018), OE (Hendrycks et al., 2019b), and Gram (Sastry & Oore, 2020). Since Lee et al. (2018) utilizes a subset of the $\mathcal{D}_{out}^{test}$ data for tuning the detection procedure while our method and OE do not, we modify Mahalanobis to tune with $\mathcal{D}_{out}^{train}$ for fair comparison. For detailed descriptions of the datasets and the experiments, refer to Supplementary E.1 and E.2.

| $\mathcal{D}_{in}$ | $\mathcal{D}_{out}^{test}$ | FPR95 ↓ | | | | AUROC ↑ | | | | AUPR ↑ | | | |
| | | Mahalanobis | OE | Gram | Ours | Mahalanobis | OE | Gram | Ours | Mahalanobis | OE | Gram | Ours |
| --- | --- | --- | --- | --- | --- | --- | --- | --- | --- | --- | --- | --- | --- |
| | Gaussian | **0.00** | 0.52 | 0.02 | 0.38 | **100.00** | 99.78 | 99.99 | 99.91 | **100.00** | 99.39 | 99.98 | 99.88 |
| | SVHN | 15.38 | 2.26 | **0.74** | 2.84 | 97.06 | 99.25 | **99.77** | 99.28 | 97.06 | 98.96 | **99.88** | 99.12 |
| CIFAR-10 | CIFAR-100 | 78.20 | 24.65 | 28.47 | **21.27** | 72.43 | 94.34 | 93.73 | **95.06** | 71.66 | 94.06 | 93.77 | **94.74** |
| | TinyImageNet | 76.11 | 31.28 | 33.71 | **27.22** | 74.26 | 94.04 | 93.56 | **94.55** | 72.21 | 94.33 | 94.02 | **94.75** |
| | LSUN | 59.61 | 9.46 | 10.15 | **7.05** | 81.40 | 97.99 | 97.56 | **98.33** | 77.08 | 97.70 | 96.81 | **98.06** |
| Average | | 27.35 | 13.63 | 14.62 | **11.75** | 91.28 | 97.08 | 96.92 | **97.43** | 90.47 | 96.87 | 96.92 | **97.31** |

Table 4: Out-of-distribution detection results. All results are percentages and averaged over 10 runs.

Table 4 shows the evaluation results. While Mahalanobis and Gram works slightly better on synthetic datasets such as Gaussian noise, on more near-distribution outliers such as CIFAR-100, TinyImageNet, and LSUN, our method outperforms these baselines by a large margin, which leads to a gain in overall performance. Our method also outperforms OE on most metrics including the Gaussian noise.

## 5 CONCLUSION

Parting from recent studies on adversarial examples, we present a new adversarial framework where the defender preemptively modifies classifier inputs. We introduce a novel optimization algorithm for finding safe spots in the vicinity of original inputs as well as a new network training method suited for enhancing preemptive robustness. The experiments show that our algorithm can find safe spots for robust classifiers on most of the correctly classified images. Further results show that they can be used to improve empirical and certified robustness on smooth classifiers. Finally, we combine the new network training scheme and the safe spot generation method to devise a new out-of-distribution detection algorithm that achieves the state of the art performance on near-distribution outliers.

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
