# OpenReview forum: "Exploiting Safe Spots in Neural Networks for Preemptive Robustness and Out-of-Distribution Detection"
_ICLR.cc/2021/Conference — Reject_

### Official Review · AnonReviewer3 · 2020-10-22
**Interesting new idea for mitigating adversarial attacks, apparently effective, but possibly difficult to apply**

**Rating:** 7
**Confidence:** 3

**Review:**

Thank you for your answers.

------

The paper proposes a new method for making adversarial attacks more difficult.
In their method, the defender (not the attacker) modifies the original input $x_o$ to $x_s$ which is guaranteed to be safe in the sense that an attacker modifying $x_s$ will not manage to change the predicted class until large changes to the sample are performed.
The defender's budget for modifying the sample is denoted $\delta$, whereas the attackers budget is $\epsilon$.
After some relaxations, they arrive at the optimization problem stated in Equation (2):
Find the modification $x_s$ (subject to budget $\delta$) which minimizes the risk (measured by cross-entropy) that any modification of $x_s$ by an attacker (subject to budget $\epsilon$) is misclassified.
They also extend the idea to out-of distribution (OOD) detection, though the main contribution seems to be in mitigating adversarial attacks during testing.

Strong Points:

- the basic idea and derivation of the optimization problem is clearly written.
- the idea of modifying an input image before classification is interesting, apparently new, and effective in mitigating the impact of adverserial attacks (according to the experiments in 4.1, 4.2, 4.3).


Weak/Unclear Points:

- Section 3.5 "SAFE SPOT-AWARE ADVERSARIAL TRAINING" is a little bit unclear to me.
Is it that now the training data, not the test data is modified? But then it is not so clear to me, whether the final training objective is still well defined.
It might just have a similar effect as adding noise to training samples.
Furthermore, it appears that the whole thing becomes difficult to train, since during training it necessary to iterate between (A) ordinary model training and (B) modification of training samples.

- Section 3.6 "OUT-OF-DISTRIBUTION DETECTION" is not convincing:
Apparently the proposed method is not only hard to train, but also has three important hyper-parameters $\gamma$, $\lambda$, and $\mu$, which need to be carefully tuned. Therefore, even though the authors report improvements over previous methods in Section 4, I am not convinced that this is a practical approach to OOD.

- In Section 4.1, I am not sure what the authors mean with "our methods can find safe spots on over 85% of the
test set images". My understanding is that, if the class label could not be changed by an attacker, then the method was successful, even if the original sample was misclassified.
However, Table 1 reports only classification accuracy.

---

> ### Author Response · Authors · 2020-11-20
> **Response to AnonReviewer3**
>
> We thank the reviewer for the encouraging comments (“the basic idea and derivation ... is clearly written”, “interesting, apparently new, and effective”) and constructive feedback. We would like to address the reviewer’s concerns below.
>
>
> **Q: “Section 3.5 "SAFE SPOT-AWARE ADVERSARIAL TRAINING" is a little bit unclear to me.”**
>
> A: Safe spot-aware adversarial training directly aims to improve preemptive robustness by simulating the safe spot generation and the following adversarial attack on the training set. The final training objective directly relates to Equation 1, while ground-truth labels are used instead as we know them during training. The mentioned iteration between ordinary model training and modification of training samples is a feature that also exists in standard adversarial training. The modification procedure does get a bit more complicated due to the safe spot search. However, as we approximate it with targeted FGSM or k-step PGD, the overall complexity does not increase as much, and can be adjusted to fit the needs of the practitioner. We have revised Section 3.5 of our paper to make this more clearer.
>
>
> **Q: “Section 3.6 "OUT-OF-DISTRIBUTION DETECTION" is not convincing: Apparently the proposed method is not only hard to train, but also has three important hyper-parameters $\gamma$, $\lambda$, and $\mu$, which need to be carefully tuned.”**
>
> A: As we describe in Supplementary E.2, the hyperparameters for OE fine-tuning, $\gamma$ and $\lambda$, do not need to be carefully tuned. We conducted a hyperparameter search for $\gamma$ and $\lambda$ on a coarse $3 \times 3$ grid. Also, the hyperparameter for the score function, $\mu$, can be easily tuned since it does not require network training.
>
>
> **Q: “In Section 4.1, I am not sure what the authors mean with "our methods can find safe spots on over 85% of the test set images”**
>
> A: Thank you for your comment. We revised our paper to make the term “safe spot” more consistent throughout the paper. In the revised version, a safe spot also has to preserve the ground-truth label of the original image. With this definition, reporting classification accuracy (under adversarial attack) is an appropriate measure of performance.

---

### Official Review · AnonReviewer4 · 2020-10-24
**A new adversarial framework for adversarial robustness**

**Rating:** 6
**Confidence:** 5

**Review:**

This paper proposes a new adversarial framework where the defender could preemptively modify classifier inputs to find safe spots that are robust to adversarial attacks. They then introduce a novel bi-level optimization algorithm that can find safe spots on over 90% of the correctly classified images for adversarially trained classifiers on CIFAR-10 and ImageNet datasets and show that they can be used to improve both the empirical and certified robustness on smoothed classifiers. Besides, they propose a new training scheme based on their conjecture about safe spots for out-of-distribution detection which achieves state-of-the-art results on near-distribution outliers.

Overall, the writing is clear and the idea is interesting. I think they have the following contributions:

1. Propose a novel adversarial framework and motivate it by a real-world application (search engine example);

2. Propose an effective algorithm to find safe spots;

3. Show the usefulness of safe spots for adversarial robustness and out-of-distribution detection.

However, I have the following concerns:

1. I am wondering whether the safe spots actually exist. Based on the definition, the classifier should have the same predictions on all data points in the $\epsilon$-ball around a safe spot. So the classifier has certified robustness on the safe spots. But it might be hard to show that the classifier has certified robustness on the safe spots. Although they have shown that the attacks could not successfully find adversarial examples for the safe spots, it might be due to the “gradient masking” issues. Could the authors try the auto-attack proposed in [1] to see whether the classifier is actually robust on the safe spots?

2. From their results, we can see that the safe spots don’t exist for naturally trained models. We need to use adversarial training to produce safe spots. But it is not surprising that adversarial training could produce safe spots. In fact, if we could solve the standard adversarial training objective optimally, then any natural images from the training distribution should be safe spots. Could the authors explain why we need to find safe spots other than the natural images in this case?

3. If the classifier is not robust on the natural image $x_o$, and the defender finds a safe spot $x_s$ around $x_o$, then from the attacker perspective, why he could not first perturb $x_s$ to be $x_o$, and then perturb $x_o$ to find adversarial examples? If the attacker could not find adversarial examples for $x_s$, then he may try other attack strategies like using larger $\epsilon$ or other perturbation types. In such cases, the proposed defense framework may not work. Could the authors explain it?

4. For safe spot-aware adversarial training, they mention that the training procedure is more computationally demanding than PGD adversarial training. Then they use targeted FGSM or k-step PGD towards the ground-truth label as a proxy to safe spot search. It is hard for me to understand what they exactly do in this part. Could the authors describe it in detail? Also, why would the safe spot-aware adversarial training be better than the standard adversarial training? I think standard adversarial training can also produce safe spots. Is it because the safe spot-aware adversarial training search for $x_a$ in a larger ball around $x_o$ ($B_{\delta+\epsilon}(x_o)$) than the standard adversarial training? Could the authors try standard adversarial training with a perturbation budget of $\delta+\epsilon$ to see if this is the case?

5. For out-of-distribution detection, they conjecture that the samples from the learned distribution will have a higher probability of having safe spots compared to the out-of-distribution samples. But I don’t think their results could support this conjecture. In their training objective, they explicitly minimize the probability of safe spot existence of the outlier samples. So they try to train a model such that their conjecture holds. It would be better if they could show whether their conjecture holds for naturally trained models or the models trained using outlier exposure. I suggest they perform an ablation study for objective (4). Also, I think they miss some OOD detection baselines, such as [2] and [3]. Could the authors compare their method to them?


[1] Croce, Francesco, and Matthias Hein. "Reliable evaluation of adversarial robustness with an ensemble of diverse parameter-free attacks." arXiv preprint arXiv:2003.01690 (2020).

[2] Mohseni, Sina, et al. "Self-Supervised Learning for Generalizable Out-of-Distribution Detection." AAAI. 2020.

[3] Liu, Weitang, et al. "Energy-based Out-of-distribution Detection." arXiv preprint arXiv:2010.03759 (2020).

--------AFTER DISCUSSION WITH AUTHORS---------

Thanks for the clarification. Some of my concerns have been addressed and I have raised my score. But I keep the concern that the proposed defense framework may be easily broken in practice given that the attacker can have unlimited power.

---

> ### Author Response · Authors · 2020-11-20
> **Response to AnonReviewer4 (1/3)**
>
> We thank the reviewer for the encouraging comments (“the writing is clear”, “the idea is interesting”, “a novel adversarial framework”, “show the usefulness of safe spots”) and constructive feedback. We would like to address the reviewer’s questions below.
>
> **Q: “I am wondering whether the safe spots actually exist. Could the authors try the auto-attack proposed in [1]”**
>
> A: We evaluated our method against auto-attack using the standard setting on CIFAR-10 $\ell_2$ threat model ($\delta=0.5$, $\epsilon=0.5$), and provide the adversarial accuracies below:
>
>
> Method	    | clean    | PGD   | AutoAttack
> ---|---|---|---
> Madry [4]           | 90.26   |  68.16  | 67.86
> Ours                  | **90.92**   |  **90.60**  | **89.57**
>
> Here, ‘Madry’ refers to measuring the original image’s robustness on a PGD adversarially trained model, and ‘Ours’ refers to measuring the robustness of our safe spot solutions from S-Full on the safe spot-aware adversarially trained model (S-FGSM+ADV). The results show that our safe spot solutions still remain robust against the stronger adversary, maintaining the adversarial accuracy gap above 20\%p.
>
> Also, as mentioned in Section 4.3 and Supplementary D.2, we applied our method to randomized smoothing, which provides a theoretical guarantee for $\ell_2$ robustness. We found that safe spots actually exist for 92% of the correctly classified images on a smoothed Gaussian model in CIFAR-10 test set.
>
>
> **Q: “if we could solve the standard adversarial training objective optimally, then any natural images from the training distribution should be safe spots.”**
>
> A: We agree that if there is an optimal classifier that is accurate and robust, we do not need safe spots. However, recent works show that in reality, there exists an inherent trade-off between adversarial robustness and standard generalization [5, 6]. Therefore, achieving both high robustness and high clean accuracy on natural images is extremely difficult. However, our safe spot-aware adversarial training, combined with our safe spot algorithm, can achieve high robustness with minimal clean accuracy drop. For example, in Table 2 (left) of the main paper, Fast adversarial training, which is a variant of standard adversarial training, only achieves an adversarial accuracy of 30% while the clean accuracy drops by 13%p compared to the natural classifier. Our safe spot-aware classifier, S-FGSM+FAST, can achieve an adversarial accuracy of 62%, which is more than twice compared to Fast adversarial training, with only a 6%p drop in clean accuracy.
>
> (continued)

---

> > ### Author Response · Authors · 2020-11-20
> > **Response to AnonReviewer4 (2/3)**
> >
> > **Q: “Why could he not first perturb $x_s$ to be $x_o$, and then perturb $x_o$ to find adversarial examples? He may try other attack strategies like using larger $\epsilon$ or other perturbation types.”**
> >
> > A: Thank you for the suggestion. We tried reverse engineering the original images from their safe spots by solving the following problem:
> >
> > $$ \\max_{x_r}  \\sup_{x_a} ~ \\ell(x_a, c(x_s))  \\:\\:  \\mathrm{subject ~ to}  \\:\\: \\| x_r - x_s \\|_p \\le \\delta \\:\\: \\mathrm{and} \\:\\: \\| x_a - x_r \\|_p \le \epsilon,$$
> >
> > where $x_r$ denotes the reconstructed image. We then generated an adversarial example $x_a$ from $x_r$ with a 20-step PGD adversarial attack varying the modification budget from a small value up to $\epsilon$, considering the case where $x_r$ is not perfectly restored to $x_o$. Since we first try to reconstruct $x_o$, we define the attack is successful if the resulting adversarial example is wrongly classified and the example lies in $B_\epsilon(x_o)$.  We conducted an experiment on CIFAR-10 $\ell_2$ threat model ($\delta=0.5$, $\epsilon=0.5$) and report the adversarial accuracies below:
> >
> > Method      \   Budget    | 0.1       | 0.2       | 0.3       | 0.4       | 0.5
> > ---|---|---|---|---|---
> > Madry [4]  |- |- | - |- | 68.16
> > Ours         |  86.21  | 81.45    | 78.84  | 79.02   | 80.23
> >
> > ‘Madry’ refers to measuring the original image’s robustness on a PGD adversarially trained model under standard PGD adversarial attack. ‘Ours’ refers to measuring the robustness of safe spots on our safe spot-aware adversarially trained model (S-FGSM+ADV) under the safe spot-aware attack devised above. The results show that our method is about 11%p more robust compared to the baseline.
> >
> > We also tried perturbing $x_s$ directly, with varying the modification budget from $\epsilon$ to $\delta + \epsilon$. Note that if the modification budget is equal to $\delta + \epsilon$, then the optimal adversarial example generated from this attack is the same as the optimal example from the attack proposed above. Even so, this attack is much easier to optimize since it does not have to solve a bi-level optimization problem. As from above, we define the attack is successful if the resulting adversarial example is wrongly classified and the perturbed example lies in $B_\epsilon(x_o)$. We conducted an experiment on CIFAR-10 $\ell_2$ threat model ($\delta=0.5$, $\epsilon=0.5$) and present the adversarial accuracies below:
> >
> > Method	  \   Budget    | 0.5       | 0.6       | 0.7       | 0.8       | 0.9       | 1.0
> > ---|---|---|---|---|---|---
> > Madry [4]             	| 68.16   |     -       |     -       |      -      |     -       |   -
> > Ours               	| 90.48   | 84.16   | 80.66   | 84.60   | 89.42   | 90.92
> >
> > ‘Madry’ refers to perturbing the original image with a standard PGD attack ($\epsilon=0.5$) on a PGD adversarially trained model. ‘Ours’ refers to perturbing the safe spot solutions up to $\delta+\epsilon$ on our safe spot-aware model (S-FGSM+ADV), also with a PGD adversarial attack. We find that our method remains more robust compared to the baseline on all the perturbation budgets, with the minimum adversarial accuracy gap being 12.5%p when the modification budget for Ours is 0.7.
> >
> > (continued)

---

> > > ### Author Response · Authors · 2020-11-20
> > > **Response to AnonReviewer4 (3/3)**
> > >
> > > **Q: “they use targeted FGSM or PGD towards the gt label as a proxy to safe spot search. It is hard for me to understand what they exactly do ... try standard adversarial training with a perturbation budget of $\delta+\epsilon$ to see if this is the case?”**
> > >
> > > A: The targeted FGSM or k-step PGD towards the ground-truth label is used as a more efficient and less accurate version of our safe spot search. Concretely, we solve the following optimization problem instead:
> > >
> > > $$\\mathop{\\rm minimize}_{x_s} \\: \\ell(x_s,c(x_o)) \\:\\: \\mathrm{subject~to} \\:\\: \\|x_s-x_o\\|_p\\le\\delta.$$
> > >
> > > In other words, we replace the inner maximization $\sup_{x_a} \ell(x_a, c(x_o))$ in the original optimization problem, which S-Full solves, by $\ell(x_s, c(x_o))$. To solve this simplified version, FGSM or PGD is applied in the direction of decreasing the loss $\ell(x_s, c(x_o))$. Using this, the safe spot-aware model simulates our defend-first-attack-later scenario while training, which results in better preemptive robustness. We have revised our paper to clarify this and the reviewer can find the detailed explanation in the last paragraph of Section 3.5.
> > >
> > > That said, we tried standard adversarial training with a perturbation budget of $\delta+\epsilon$ on CIFAR-10 $\ell_2$ threat model ($\epsilon=0.5$, $\delta=0.5$) and present the clean and adversarial accuracies below:
> > >
> > > Method|clean|PGD
> > > -|-|-
> > > Madry [4]|90.26|68.16
> > > Madry (budget=1.0) [4]|84.50|66.51
> > > Ours|**90.92**|**90.60**
> > >
> > > ‘Madry’ and ‘Madry (budget=1.0)’ refer to measuring the original image’s robustness on a PGD adversarially trained model, trained with budget=0.5 and budget=1.0, respectively. ‘Ours’ refers to measuring the robustness of the safe spot solutions on our safe spot-aware adversarially trained model (S-FGSM+ADV). PGD adversarial attack was used to measure the adversarial accuracies. We observe that our method outperforms both baselines by a large margin (about 22%p). Also, we can find that ‘Madry (budget=1.0)’, while trained with a stronger attack, is actually less robust than the others on a smaller perturbation budget.
> > >
> > > **Q: “For out-of-distribution detection, they conjecture that the samples from the learned distribution will have a higher probability of having safe spots compared to the out-of-distribution samples. But I don’t think their results could support this conjecture.”**
> > >
> > > A: Figure 3 plots the loss histograms using a safe spot-aware model, which was not trained to satisfy the conjecture. The results show that the out-of-distribution samples have much less probability of having safe spots than the in-distribution samples, as shown in the right figure. This shows that our conjecture on the likeliness of safe spot existence holds on models that are not trained for the conjecture. We also made this clearer on the revised version of our paper.
> > >
> > > **Q: “they miss some OOD detection baselines, such as [2] and [3]”**
> > >
> > > A: We compared our method to [2, 3]. For the implementation of [2], note that [2] fine-tunes a network with OE for 100 epochs, while other OE-based methods do for 10 epochs. For a fair comparison, we restricted the number of epochs for fine-tuning to 10. Also, we conducted the hyperparameter search for the weight $\lambda \in \\{0.5, 1.0, 2.0\\}$. For the implementation of [3], we set the weight $\lambda$ to 0.1 and conducted a grid search for choosing the optimal hyperparameters $m_{\text{in}} \in \\{-15, -19, -23, -27\\}$ and $m_{\text{out}} \in \\{-3, -5, -7\\}$ that minimize FPR95, following the experimental protocol in [3]. Also, we updated our OOD results by tuning the learning rate $\beta$ for safe spot searching. The results below show that our method still outperforms [2, 3] in AUROC and AUPR.
> > >
> > >
> > > OOD|FPR95$\downarrow$|AUROC$\uparrow$|AUPR $\uparrow$
> > > -|-|-|-
> > > Gaussian|1.03 / 1.80 / **0.38**|99.16 / 99.10 / **99.91**|96.37 / 97.61 / **99.88**
> > > SVHN|**1.71** / 3.97 / 2.84|99.27 / 97.38 / **99.28**|98.50 / 92.94 / **99.12**
> > > CIFAR100|22.52 / **17.93** / 21.27|94.79 / 94.62 / **95.06**|94.71 / 92.23 / **94.74**
> > > TinyImageNet|27.73 / **22.17** / 27.22|94.52 / 94.42 / **94.55**|**94.78** / 92.53 / 94.75
> > > LSUN |6.72 / **4.68** / 7.05|98.23 / 97.44 / **98.33**|97.92 / 94.53 / **98.06**
> > > Average|11.94 / **10.11** / 11.75|97.19 / 96.59 / **97.43**|96.46 / 93.97 / **97.31**
> > >
> > > **Table**: OOD detection results with additional baselines ([2] / [3] / ours).
> > >
> > >
> > > [1] Croce, et.al., Reliable evaluation of adversarial robustness with an ensemble of diverse parameter-free attacks, ICML 2020.
> > >
> > > [2] Mohseni, et. al., Self-Supervised Learning for Generalizable Out-of-Distribution Detection, AAAI 2020.
> > >
> > > [3] Liu, et. al., Energy-based Out-of-distribution Detection, NeurIPS 2020.
> > >
> > > [4] Madry, et. al., Towards Deep Learning Models Resistant to Adversarial Attacks, ICLR 2018.
> > >
> > > [5] Tsipras, et al., Robustness bay be at odds with accuracy, ICLR 2019.
> > >
> > > [6] Zhang, et al., Theoretically principled trade-off between robustness and accuracy, ICML 2019.

---

> ### Comment · AnonReviewer4 · 2020-11-20
> **Some concerns remain**
>
> Thanks for the clarification. Some of my concerns have been addressed. But I still have some concerns:
>
> 1. Could the authors also show CIFAR-10 $\ell_\infty$ threat model results under auto-attack?
>
> 2. The authors' response doesn't address my concern about why we don't just use standard adversarial training to produce safe spots? The issue that adversarial training will affect clean accuracy is well known and some techniques [1] have been proposed to mitigate this problem. But it seems not related to my question.
>
> 3. I still have concerns about why the attackers don't just try a larger perturbation budget. If there exist adversarial examples near $x_o$ and the safe spot $x_s$ is close to $x_o$, then we should be able to find adversarial examples near $x_s$ with other attack strategies like using a larger perturbation budget. The experiments provided cannot convince me that such kinds of attacks won't exist. There might be other attack methods that could break the proposed defense framework. We don't want to have a defense approach that may be broken easily later. I think the authors can refer to [2] for how to evaluate adversarial robustness. Also, I think the criterion to measure the attack's success may not be proper: why the resulting adversarial example should lie in $B_\epsilon(x_o)$? I think it can also lie in $B_{\epsilon+\delta}(x_o)$ since $x_a$ may be in $B_{\epsilon+\delta}(x_o)$.
>
> 4. Could you plot the same figure as Figure 3 for CIFAR-10 vs SVHN? I want to see if the conjecture still holds for the harder OOD datasets.
>
>
> [1] Xie, Cihang, et al. "Adversarial examples improve image recognition." Proceedings of the IEEE/CVF Conference on Computer Vision and Pattern Recognition. 2020.
>
> [2] Carlini, Nicholas, et al. "On evaluating adversarial robustness." arXiv preprint arXiv:1902.06705 (2019).

---

> > ### Author Response · Authors · 2020-11-21
> > **Response to AnonReviewer4 (1/2)**
> >
> > Thank you for the quick response. We address the reviewer’s concerns below.
> >
> > **Q1. “Could the authors also show CIFAR-10 $\ell_\infty$ threat model results under auto-attack?”**
> >
> > A1. We additionally evaluated our method against auto-attack using the standard setting on CIFAR-10 $\ell_\infty$ threat model ($\delta=8/255$, $\epsilon=8/255$), and provide the adversarial accuracies below:
> >
> > Method| clean |  PGD |AutoAttack
> > -|-|-|-
> > Madry | 86.51   |  47.21  | 46.24
> > Ours  | **86.83**   |  **85.35**  | **81.74**
> >
> > Here, ‘Madry’ refers to measuring the original image’s robustness on a PGD adversarially trained model, and ‘Ours’ refers to measuring the robustness of our safe spot solutions from S-Full on the safe spot-aware adversarially trained model (S-FGSM+ADV). The results show that our safe spot solutions still remain robust against the stronger adversary, maintaining the adversarial accuracy gap above 35\%p.
> >
> > **Q2. “The authors' response doesn't address my concern about why we don't just use standard adversarial training to produce safe spots?”**
> >
> > A2: We understood the reviewer’s concern as “If we solve the standard adversarial training objective optimally, such that the trained model achieves adversarial accuracy on par with the clean accuracy of naturally trained models, most of the natural images will already be safe spots.” Our claim in the last response was that while the reviewer’s statement is correct, it is extremely hard to build an optimally robust classifier that achieves such performance in practice.
> >
> > One major difficulty in achieving high adversarial accuracy is that even if standard adversarial training turns every ‘training’ datapoint to a safe spot, this does not translate to every ‘test’ datapoint becoming a safe spot. The theoretical work of [2] shows that adversarial training by nature suffers from a significant generalization gap as its sample complexity is substantially larger than standard training. The authors also conduct extensive experiments and conclude that it is much harder to train a robust model with a certain level of test adversarial accuracy than training a natural model with the same level of test clean accuracy. As noted from the previous response, other works [4, 5] also show that there exists an inherent trade-off between adversarial robustness and standard generalization, which further complicates the problem.
> >
> > Our work can be thought of as a novel attempt to tackle the problems above by preemptively manipulating natural images, unlike other existing methods relying only on network training, improving adversarial robustness with minimal clean accuracy drop.
> >
> > We thank the reviewer for pointing us to a valuable work ([1]), but this work seems to aim at improving standard generalization, not adversarial robustness. We evaluated the test adversarial accuracy of an EfficientNet-B4 + AdvProp model against $\ell_\infty$ PGD of size $\epsilon=4/255$, which is the same attack used for training. The model achieves 1.44% adversarial accuracy, which is far less than standard adversarial training’s 32.57% (see Table 2 (left)). Therefore, it is hard for us to conclude whether the work mitigates the issues raised above or not.
> >
> > If the reviewer were mentioning something else, we would be grateful if the reviewer could elaborate on the question once more.
> >
> > (continued)

---

> > > ### Author Response · Authors · 2020-11-21
> > > **Response to AnonReviewer4 (2/2, Q3 results updated)**
> > >
> > > **Q3. “I still have concerns about why the attackers don't just try a larger perturbation budget.”**
> > >
> > > We agree with the reviewer that attackers may increase their perturbation budget to make their attack stronger, and hence it is crucial for a defense method to maintain its robustness to a certain level when the perturbation budget is increased. When we focused on a particular $\epsilon$ ball until the previous response, our intention was 1) to provide a more direct comparison to other defense methods and 2) to ensure that the perturbed image does not stray too far from the safe spot. While it is true that an attacker can generate more successful adversarial examples when the perturbation budget is increased, this also risks the attacker that the perturbed example may also be easily noticed by our visual system, which apparently goes against his/her purpose (see Supplementary F.5 of [3]).
> > >
> > > Particularly in our defense method, as the reviewer mentioned, if there exists an adversarial example $x_a \in B_{\epsilon}(x_o)$ near a natural image $x_o$ and the safe spot $x_s \in B_{\delta}(x_o)$ is close to $x_o$, then $x_a \in B_{\delta + \epsilon}(x_s)$ and an adversary can find an adversarial example for $x_s$ by increasing the perturbation budget to $\delta + \epsilon$. Nevertheless, the attacker may need to spend all its increased budget ($\delta+\epsilon$) in the worst case, which is troublesome for the attacker, as noted above.
> > >
> > > That said, to evaluate our defense method against larger perturbations, we ran additional experiments by increasing the perturbation budget from $\epsilon$ to $\delta+\epsilon$ on CIFAR-10 $\ell_2$ threat model ($\epsilon=0.5$, $\delta=0.5$) and present the results below.
> > >
> > >
> > > Method  \   Budget    | 0.5       | 0.6       | 0.7       | 0.8       | 0.9       | 1.0
> > > -|-|-|-|-|-|-
> > > Madry [4]             	| 67.86   | 61.83   | 55.11   | 47.72   | 40.49   | 33.93
> > > Ours                         | **89.57**   | **83.19**   | **76.41**   | **66.27**   | **53.01**   | **36.76**
> > >
> > >
> > > Here, ‘Madry’ refers to measuring the original image’s robustness on a PGD adversarially trained model, and ‘Ours’ refer to measuring the robustness of our safe spot solutions from S-Full on the safe spot-aware adversarially trained model (S-FGSM+ADV), where we set the defender to use $\epsilon=1.0$ when generating the safe spots ($\delta$ is fixed to $0.5$). All images were perturbed with AutoAttack. The results show that our safe spot solutions still remain robust against the stronger adversary on all perturbation budgets, compared to the baseline. Also, the performance gap is above 10%p on all budget sizes except 1.0.
> > >
> > > We thank the reviewer for the remark and will include the discussions above in our future revision.
> > >
> > >
> > > **Q4. “Could you plot the same figure as Figure 3 for CIFAR-10 vs SVHN?”**
> > >
> > > We appended the additional plots for CIFAR-10 vs. (SVHN, CIFAR-100, TinyImageNet) in Figure 2 in Supplementary Material. The FPR95 scores show that our conjecture also holds for other OOD datasets.
> > >
> > > [1] Xie, et. al., Adversarial examples improve image recognition, CVPR 2020.
> > >
> > > [2] Schmidt, et. al., Adversarially Robust Generalization Requires More Data. NeurIPS 2018.
> > >
> > > [3] Qin, et. al., Adversarial Robustness through Local Linearization, NeurIPS 2019.
> > >
> > > [4] Tsipras, et al., Robustness bay be at odds with accuracy, ICLR 2019.
> > >
> > > [5] Zhang, et al., Theoretically principled trade-off between robustness and accuracy, ICML 2019.

---

> > > > ### Comment · AnonReviewer4 · 2020-11-23
> > > > **Primary concern**
> > > >
> > > > Thanks for the results. My primary concern is still that the attacker may try other attack strategies to find adversarial examples near the safe spots. As I mentioned, since there exists an adversarial example $x_a$ near a natural image $x_o$, and the safe spot $x_s$ is close to the natural image $x_o$, then the attacker should be able to find the adversarial example $x_a$.  Also, by the definition of adversarial example, $x_a$ should be similar to $x_o$, which means it could not be easily noticed by our visual system.
> > > >
> > > > Based on the authors' response, I cannot be convinced that the attacker could not find such $x_a$. From the current experimental results, it seems such $x_a$ indeed exists when the perturbation budget is $\delta+\epsilon$. Also, the authors should think about whether there exist other efficient attack algorithms that could find such $x_a$. I think the authors need to have more discussions about why finding such $x_a$ is hard and provide more experimental results to support it.
> > > >
> > > > I keep the same rating due to this primary concern.

---

> ### Author Response · Authors · 2020-11-24
> **Response to the primary concern**
>
> We thank the reviewer for the constructive response.
>
> However, the reviewer seems to have misunderstood our experiment results. The reviewer claims two things, that 1) our method can be broken when the attackers find an adversarial example $x_a$ close to $x_o$, which is true, and that 2) there should be various ways to achieve it, which has to be shown. The experiment results in our response comments show that the second claim may not hold for attackers who cannot access $x_o$.
>
> Let’s assume the attacker increases its perturbation budget to $\delta+\epsilon$. It is obvious that if there exists a misclassified example $x_a \in B_{\epsilon}(x_o)$, then $x_a \in B_{\delta+\epsilon}(x_s)$ by the definition of safe spot. However, this does not imply that a misclassified example $x_a’$ of $x_s$ generated by the attacker necessarily lies in $B_{\epsilon}(x_o)$. This discrepancy is troublesome for the attacker as the resulting $x_a’$ that is outside $B_{\epsilon}(x_o)$ may be easily detected by our visual system, as we mentioned. If the attacker has some information about the original $x_o$, then he/she may induce $x_a’$ to be also close to $x_o$ by simply adding the distance term $\\| x_a’ - x_o \\|$ to the objective. However, without knowing $x_o$, there is no straightforward way to compute the distance term or make $x_a’$ lie in $B_{\epsilon}(x_o)$.
>
> Therefore, in our previous response, “Response to AnonReviewer4 (2/3)”, we devised two strategies to find an adversarial example $x_a$ that is close to $x_o$ using only $x_s$. The first strategy was to reconstruct $x_o$ from $x_s$ by reverse engineering, and perturb the reconstructed image $x_r$ within $B_{\epsilon}(x_r)$. The second strategy was to directly find $x_a$ by perturbing $x_s$ up to $\delta+\epsilon$. In both the cases, we rejected perturbed examples that lie outside $B_{\epsilon}(x_o)$, since proper adversarial examples should be similar to $x_o$, as noted above. The experiment results show that the two attacks above cannot reliably find $x_a$ within $B_\epsilon(x_o)$.
>
> The reviewer notes that “From the current experimental results, it seems such $x_a$ indeed exists when the perturbation budget is $\delta+\epsilon$”, which is probably referring to the experiment result in “Response to AnonReviewer4 (2/2, Q3 results updated)”. However, in this experiment, we measured the adversarial accuracy without considering the distance to $x_o$. The experiment was to evaluate the performance of our method when the perturbation budget increases, as the reviewer asked “why the attackers don’t just try a larger perturbation budget”. It was not intended to show that we can find an adversarial example $x_a \in B_\epsilon (x_o)$ by increasing the budget.
>
> We agree that still, there might be other candidate attack strategies that can be tried. However, the two attacks above were not able to sufficiently penetrate our defense method.

---

> > ### Comment · AnonReviewer4 · 2020-11-24
> > **Thanks for the clarification, but the primary concern remains**
> >
> > Thanks for the clarification!
> >
> > I think it might not be reasonable to require $x_a\in B_\epsilon(x_o)$. As long as $x_a$ is visually similar to $x_o$ or $x_s$, it should be acceptable. And in practice, the attacker can manually check if $x_a$ is visually different from $x_s$ or not. My primary concern is that the proposed defense framework may be easily broken in practice given that the attacker can have unlimited power.
> >
> > Also, even if we assume that $x_a$ should be in $B_\epsilon(x_o)$, the current results still cannot convince me that we cannot successfully find $x_a$ since the attacks tried may not be strong enough. I think one attack strategy can be: the attacker first randomly perturbs $x_s$ in $B_\delta(x_s)$ to get $x_o'$ and then find adversarial examples in $B_\epsilon(x_o')$. We can repeat this process many times until we successfully find $x_a$ that is in $B_\epsilon(x_o)$. In practice, this is doable since the attacker can manually check (or run some algorithms to check) if the resulting $x_a$ is visually similar to $x_s$ (and hence similar to $x_o$).

---

> > > ### Author Response · Authors · 2020-11-25
> > > **Additional response to the primary concern**
> > >
> > > Thank you for the response.
> > >
> > > The reviewer mentions that an adversarial example $x_a$ does not have to lie in $B_\epsilon (x_o)$ in order for it to be visually similar to $x_o$. We agree that there seems to be a discrepancy between $\ell_p$ distance and visual similarity, as suggested in other works [1, 2, 3]. That said, we adopted $\ell_p$ distance as the measure of visual similarity following the common practice in adversarial robustness literature, which is why we limited the adversarial examples (from the response experiments) to lie in $B_\epsilon (x_o)$. However, if a better visual distance metric is discovered, our method can be adjusted accordingly by replacing the $\ell_p$ distance term with the new metric.
> > >
> > > The reviewer also suggests to find $x_a \in B_\epsilon (x_o)$ by repeating the attack procedure and manually checking if the generated example is visually close to $x_s$ (or $x_o$). Our additional experiments show that finding a misclassified example $x_a \in B_\epsilon (x_o)$ by repetitively sampling perturbed examples from $B_{\delta+\epsilon} (x_s)$ is ineffective in practice in terms of computational cost and required human effort for the manual checking.
> > >
> > > We evaluated our method against our second attack strategy in "Response to AnonReviewer4 (2/3)" using 100 random restarts (we found reconstructing $x_o'$ using random perturbation and attacking $x_o'$ ineffective). Concretely, we generate $x_a$ from $x_s$ with a 20-step PGD varying the modification budget from $\epsilon=0.5$ to $\delta+\epsilon=1.0$ . We define the attack successful if there exists at least one misclassified example within $B_\epsilon (x_o)$ out of the 100 perturbed images generated by the attack. The adversarial accuracy results are shown below.
> > >
> > >
> > > Method  \   Budget    | 0.5 | 0.6       | 0.7       | 0.8       | 0.9       | 1.0
> > > ---|---|---|---|---|---|---
> > > Madry [4]  		| 68.03 | -          |  -          |   -        |  -          |
> > > Ours        		| 90.17 | 81.78  |  78.48   | 82.71  | 88.54   | 90.92
> > >
> > > ‘Ours’ refers to evaluating our safe spot solutions on our safe spot-aware model (S-FGSM+ADV) against the proposed attack method. ‘Madry’ refers to evaluating the original images on a PGD adversarially trained model against a standard PGD attack with 100 random restarts. We find that our method remains more robust compared to the baseline on all $\epsilon$ budgets, with the minimum adversarial accuracy gap above 10\%p. This result shows that even if the attacker runs the attack algorithm and manually looks into the resulting image 100 times, it is not sufficient to successfully break our defense.
> > >
> > >
> > > [1] Johnson, et. al., Perceptual Losses for Real-Time Style Transfer and Super-Resolution, ECCV 2016.
> > >
> > > [2] Isola, et. al., Image-to-image translation with conditional adversarial networks, CVPR 2017.
> > >
> > > [3] Ghiasi, et. al., Breaking certified defenses: Semantic adversarial examples with spoofed robustness certificates, ICLR 2020.

---

### Official Review · AnonReviewer1 · 2020-11-03
**interesting method to identify images that are less prone to adversarial attack**

**Rating:** 5
**Confidence:** 2

**Review:**

The authors argue that there are some safe "spots" in the data space that are less prone to adversarial attacks. The authors propose a technique to identify such "safe spots". They then leverage them for robust training and observe higher robust accuracy than baseline. Finally, they leverage this observation to identify out of distribution data.

The application is important and the results look promising. However, I have the following concern:

- The authors propose a new threat model where the adversary may have access to the labeled data. They motivated such a setting with an example of Google image search. However, such a setting is quite limited. There are also existing methods that use supervised learning setting with incorrect labeling. The paper should discuss how they differ from such a line of work.

- The search algorithm requires that a correct predicted label is available. This setting is not quite realistic. How can we find a safe spot when the label is unknown.

- Some of the findings are not quite surprising. For example, a safe spot is more in a robust model with small epsilon.

---

> ### Author Response · Authors · 2020-11-20
> **Response to AnonReviewer1**
>
> We thank the reviewer for the encouraging comments (“The application is important”, “the results look promising”) and constructive feedback. We would like to address the reviewer’s concerns below.
>
> **Q: “The authors propose a new threat model where the adversary may have access to the labeled data. However, such a setting is quite limited.”**
>
> A: Concretely, there are three adversaries in our framework. The first is the virtual adversary (A) used by the defender while training a safe-spot aware classifier. The second is the virtual adversary (B) used by the defender while generating safe spots after the model training. The last adversary (C) is the real adversary that attacks the generated safe spots. In our setting, A has access to the labeled data, assuming ground-truth labels are available during training. However, B does not have access to the labeled data, because as the reviewer mentioned, random images in the real-world usually do not have annotated labels. Lastly, the real adversary C has access to the labels, which is common in adversarial attack literature, including those in real-world settings [1, 2, 3, 4].
>
>
> **Q: “There are also existing methods that use supervised learning setting with incorrect labeling. The paper should discuss how they differ from such a line of work.”**
>
> A: Our defense framework consists of two main parts. The first part is to train a safe spot-aware classifier, given a correctly labeled dataset. The second part is to find a safe spot from an unlabeled natural image, given a pre-trained classifier. Therefore, to be exact, our work is not closely related to supervised learning with incorrect labeling. However, to make our framework more applicable, we could consider a situation where the training data in the first part is wrongly labeled. Developing a safe spot-aware classifier that considers such label corruption scenario will be a good direction for future research.
>
>
> **Q: “The search algorithm requires that a correct predicted label is available. How can we find a safe spot when the label is unknown.”**
>
> A: If the reviewer meant the ground-truth label is unknown, then our search algorithm already assumes the situation where the ground-truth label is unknown. We have revised our paper to describe our safe spot searching algorithm more clearly in the second paragraph of Section 3.2.
>
> On the other hand, if the reviewer meant the predicted label may be incorrect, we note that it is not possible to find a safe spot $x_s$ for an incorrectly classified image $x_o$ if the modification budgets of the defender and the adversary are the same, *i.e.*, $\epsilon = \delta$. To see why, suppose there exists a safe spot $x_s$ of $x_o$. Since $x_o \in B_{\delta}(x_s) = B_{\epsilon}(x_s)$, it satisfies that $y_o = c(x_s) = c(x_o)$, which is directly from the definition of safe spot. However, it contradicts the fact that $x_o$ is misclassified, *i.e.*, $y_o \neq c(x_o)$.
>
>
> **Q: “Some of the findings are not quite surprising. For example, a safe spot is more in a robust model with small epsilon.”**
>
> A: It is natural to guess that more robust classifiers will also have more safe spots near the natural images, thus being more preemptively robust. However, our experiments on ImageNet in Section 4.2 show that the robust model trained with smaller $\epsilon_{\text{train}}$, which is obviously less robust than its larger $\epsilon_{\text{train}}$ counterpart, is actually more preemptively robust. This shows that the conventional notion of robustness does not necessarily translate to preemptive robustness.
>
> [1] Szegedy, et al., Intriguing properties of neural networks, arXiv:1312.6199.
>
> [2] Goodfellow, et al., Explaining and harnessing adversarial examples, ICLR 2015.
>
> [3] Kurakin, et al., Adversarial examples in the physical-world, ICLR workshop 2017.
>
> [4] Eykholt, et al., Robust Physical-World Attacks on Deep Learning Models, CVPR 2018.

---

### Official Review · AnonReviewer2 · 2020-11-10
**Look at a different perspective to improve adversarial robustness**

**Rating:** 6
**Confidence:** 4

**Review:**

Summary:
This paper aims to improve adversarial robustness of the classifiers in a different perspective than the existing works. Usually, the networks are trained using adversarial examples to improve robustness (adversarial training). This work extend this line of thought and make an input robust to adversarial attacks. Instead of updating the network, they make updates to the input to gain robustness. In other words, this work explore the existence of safe spots near the input samples that are robust against adversarial attacks. Results on CIFAR-10 and ImageNet reveals that there exists such safe spots which are resistant to adversarial perturbations and improve adversarial robustness when combined with adversarial training (the authors term it as safe-spot aware adversarial training). Based on this approach, the authors also propose out-of-distribution detection method that outperforms previous works.

Strengths:
+ Motivation is clear.
+ The proposed approach is interesting and different from existing works. The practical application of the proposed framework is elaborated clearly.
+ Technical details and formulations are clear.
+ Results show that the proposed approach improves adversarial robustness and clean data performance on both CIFAR-10 and ImageNet. Furthermore, the proposed approach greatly improves the robustness when evaluated with randomized smoothing.
+ The design of the approach enables out-of-distribution detection that outperforms previous works.

Weaknesses:
-	The major concern lies in the evaluation of the proposed technique. Here, the authors find safe spots and also propose safe-spot aware adversarial training but evaluate on PGD based adversarial attack in a standard  manner. It is important to address the possibility of safe spot aware adversarial attack on the proposed defense and its success rate. In case such attack is infeasible, please provide the rationale behind that.
-	Clarify the difference between S-Full and S-PGD from Experiments section. Since S-Full also uses T-step PGD, how it is different than S-PGD?
-	Though the out-of-distribution detection results slightly outperforms previous works under FPR95 metric, the performance gains are very minimal and not very significant than the baseline OE (Hendrycks et al., 2019b) under two metrics AUROC and AUPR.

Final thoughts:
The proposed method is clearly motivated. Although the performance gains on adversarial robustness is significant, there are critical points yet to be addressed. Therefore, I marginally accept this paper.

------------------------------------------------------------------------------------------------------------------------------------
Post rebuttal:
The authors have addressed my concerns in the rebuttal. However, I also agree with the other critical points raised by other reviewers (particularly Reviewer 4) that are of major concern. Hence, I retain my initial score and marginally accept the paper.

---

> ### Author Response · Authors · 2020-11-20
> **Response to AnonReviewer2 (1/2)**
>
> We thank the reviewer for the encouraging comments (“Motivation is clear”, “interesting and different from existing works”, “technical details and formulations are clear”) and constructive feedback. We would like to address the reviewer’s concerns below.
>
> **Q: “Address the possibility of safe spot aware adversarial attack.”**
>
> A: Most of the defense methods that are vulnerable against defense-aware adversarial attacks heavily rely on gradient masking [1, 2]. Defense methods with gradient masking typically add randomized or non-differentiable operations into the model inference or increase the model complexity significantly, which makes it hard for the adversary to generate adversarial examples by backpropagation [3, 4, 5]. Recently, it has been shown that another line of defenses that replaces the cross-entropy loss with a new loss function during training also exhibits some gradient masking [6, 7]. While these methods are resistant against gradient-based attacks with the cross-entropy loss, they fail against alternative gradient-based attacks using a different loss function.
>
> Unlike these methods above, our safe spot-aware training does not add any additional complex operations, change the network structure, or replace the cross-entropy loss for training with another loss function. Also, our safe spot algorithm only manipulates images before the adversary’s incursion and does not intervene during the attack process. Therefore, under our defense framework, the exact calculation of the loss gradient is straightforward, which means that standard gradient-based attacks can be a reliable measure to evaluate the robustness of our method.
>
> That said, we additionally evaluated the robustness of our defense framework using auto-attack [8], which contains an attack method that does not rely on gradient information. We conducted the evaluation on CIFAR-10 $\ell_2$ threat model ($\delta=0.5$, $\epsilon=0.5$) and provide the adversarial accuracies below:
>
> Method	    | Clean    | PGD   | AutoAttack
> ---|---|---|---
> Madry [9]           | 90.26   |  68.16  | 67.86
> Ours                  | **90.92**   |  **90.60**  | **89.57**
>
> Here, ‘Madry’ refers to measuring the original images’ robustness on a PGD adversarially trained model, and ‘Ours’ refers to measuring the robustness of our safe spot solutions from S-Full on the safe spot-aware adversarially trained model (S-FGSM+ADV). The results show that our safe spot solutions still remain robust against the stronger adversary, maintaining the adversarial accuracy gap above 20\%p.
>
> **Q: “Clarify the difference between S-Full and S-PGD.”**
>
> A: Concretely, S-PGD can be seen as solving the following optimization problem:
>
> $$ \\mathop{\\rm minimize}_{x_s} \\: \\ell(x_s, c(x_o)) \\:\\: \\mathrm{subject~to} \\:\\: \\| x_s - x_o \\| \\le \\delta. $$
>
> In other words, S-PGD replaces the inner maximization $\sup_{x_a} \ell(x_a, c(x_o))$ in our safe spot objective, by $\ell(x_s, c(x_o))$. To solve this simplified version, PGD is applied in the direction of decreasing the loss $\ell(x_s, c(x_o))$. We have revised our paper to clarify this difference and the reviewer can find the detailed description in the last paragraph of Section 3.5.
>
> (continued)

---

> > ### Author Response · Authors · 2020-11-20
> > **Response to AnonReviewer2 (2/2)**
> >
> > **Q: “The performance gains are very minimal and not very significant than the baseline OE.”**
> >
> > A: We first note that we tuned the learning rate $\beta$ for safe spot search and updated our OOD results. In the updated results, our method outperforms the baselines by a larger margin than before (see section 4.4). Also, we conducted one-tailed t-tests and measured the p-values to quantify the significance of the performance gains compared to the baseline OE. We show the results below.
> >
> > (Null hypothesis: our score is better than the baseline OE’s score.)
> >
> > OOD		   | FPR95 $\downarrow$ | AUROC $\uparrow$ | AUPR $\uparrow$
> > -|-|-|-
> > Gaussian         | **2.57\*10^-8**                   | **1.75\*10^-16**             | **1.94\*10^-20**
> > SVHN              | 1.0-4.18\*10^-11            | **4.97\*10^-9**               | **2.45\*10^-13**
> > CIFAR100        | **1.57\*10^-17**                 | **1.80\*10^-21**             | **3.19\*10^-19**
> > TinyImageNet  | **1.01\*10^-15**                 | **6.66\*10^-22**             | **6.59\*10^-17**
> > LSUN               | **4.53\*10^-17**                 | **6.91\*10^-19**             | **2.08\*10^-15**
> >
> > We conclude that our method is statistically significant compared to OE at a significance level of 0.01, except for the FPR95 score on SVHN.
> >
> > [1] Athalye, et. al., Obfuscated Gradients Give a False Sense of Security: Circumventing Defenses to Adversarial Examples, ICML 2018.
> >
> > [2] Tramer, et. al., On Adaptive Attacks to Adversarial Example Defenses, NeurIPS 2020.
> >
> > [3] Roth, et. al., The odds are odd: A statistical test for detecting adversarial examples, ICML 2019.
> >
> > [4] Xiao, et. al., Enhancing Adversarial Defense by k-Winners-Take-All, ICLR 2020.
> >
> > [5] Li, et. al., Are generative classifiers more robust to adversarial attacks?, ICML 2019.
> >
> > [6] Zhang, et. al., Defense Against Adversarial Attacks Using Feature Scattering-based Adversarial Training, NeurIPS 2020.
> >
> > [7] Pang, et. al., Rethinking softmax cross-entropy loss for adversarial robustness. ICLR 2020.
> >
> > [8] Croce, et. al., Reliable Evaluation of Adversarial Robustness with an Ensemble of Diverse Parameter-free Attacks, ICML 2020.
> >
> > [9] Madry, et. al., Towards Deep Learning Models Resistant to Adversarial Attacks, ICLR 2018.

---

### Author Response · Authors · 2020-11-20
**Dear reviewers**

We thank the reviewers for their comprehensive and thoughtful comments. We carefully revised our submission based on the reviews, and hope the revisions address the reviewers’ concerns.

Main revisions:
- We modified the definition of preemptive robustness to include preserving the ground-truth labels to make the usage of the related terms (*e.g.* safe spot) more consistent. (Section 3.1, 3.2)
- We specified on each of our methods whether they use the ground-truth labels or the predicted labels. We also included the rationale behind these assumptions more clearly. (Section 3.2, 3.5, 3.6)
- We added a more detailed explanation of our safe spot-aware training method. (Section 3.5)
- We tuned the learning rate $\beta$ for safe spot search on OOD detection and updated our OOD results (Section 4.4).

---

### Decision · Program_Chairs · 2021-01-07
**Final Decision**

**Decision:**

Reject

**Comment:**

The reviewers recognized that the proposed method is interesting and seems to be useful in some cases, and the authors provided sufficient empirical results to support their claim.
In addition, some comments have already been clarified.
However, some reviewers still concerned that the proposed defence method will be defeated under some conditions, and still have the major concern regarding the issue of adopting some attack strategies to find adversarial examples near the safe spots, even though the authors clarified some critical points of the proposed method.
These drawbacks led to the decision to not accept. However, this paper has some merit and can be made into a stronger contribution in the future.